# Recent Insight about HE4 Role in Ovarian Cancer Oncogenesis

**DOI:** 10.3390/ijms241310479

**Published:** 2023-06-22

**Authors:** Emanuela Anastasi, Antonella Farina, Teresa Granato, Flavia Colaiacovo, Beatrice Pucci, Sara Tartaglione, Antonio Angeloni

**Affiliations:** 1Department of Experimental Medicine, Sapienza University of Rome, Viale Regina Elena 324, 00161 Rome, Italy; emanuela.anastasi@uniroma1.it (E.A.); antoeffe22@gmail.com (A.F.); flavia.colaiacovo07@gmail.com (F.C.); beatrice.pucci94@gmail.com (B.P.); sara.tartaglione@uniroma1.it (S.T.); 2CNR-IBPM, Institute of Molecular Biology and Pathology, Department of Molecular Medicine, Sapienza University of Rome, Viale Regina Elena 324, 00161 Rome, Italy; teresa.granato@cnr.it

**Keywords:** ovarian cancer, HE4, oncogenesis, target therapy

## Abstract

Currently, ovarian cancer (OC) is a target of intense biomarkers research because of its frequent late diagnosis and poor prognosis. Serum determination of Human epididymis protein 4 (HE4) is a very important early detection test. Most interestingly, HE4 plays a unique role in OC as it has been implicated not only in OC diagnosis but also in the prognosis and recurrence of this lethal neoplasm, actually acting as a clinical biomarker. There are several evidence about the predictive power of HE4 clinically, conversely less has been described concerning its role in OC oncogenesis. Based on these considerations, the main goal of this review is to clarify the role of HE4 in OC proliferation, angiogenesis, metastatization, immune response and also in the development of targeted therapy. Through a deeper understanding of its functions as a key molecule in the oncogenetic processes underlying OC, HE4 could be possibly considered as an essential resource not only for diagnosis but also for prognosis and therapy choice.

## 1. Introduction

Ovarian cancer (OC) is the eighth most common and the fifth deadliest cancer in women worldwide with a survival rate of just 30%. Symptoms are few and nonspecific, thus about 70% of the cases are diagnosed in the advanced stages [1]. Ninety per cent of advanced ovarian cancer patients manifest malignant ascites ascribed to the progress of peritoneal carcinomatosis: progressive OC is characterized by metastases in the peritoneal cavity and-or in the retroperitoneal lymph nodes, supporting extensive disease spread beyond the abdomen [2,3].

Currently, OC is often first identified by transvaginal ultrasound (TVU) after a clinical examination. Several ultrasound features have been recognized to predict a malignancy, but the diagnostic precision still needs to be improved [4]. Serum biomarkers are an easily accessible, cost-effective, and non-invasive tool for a timely detection of OC. However, the heterogeneity of OC is a great obstacle in discovering reliable biomarkers to aid early diagnosis.

Gynecologic neoplasms comprise lesions of different origin such as epithelial, mesenchymal, sex cord-stromal, and germ cell. Of these categories, 75% is represented by high-grade serous ovarian carcinoma (HGSOC), with a poor survival [5,6]. Comparative transcriptomic and genomic studies of mouse models and organoids suggest that extracellular matrix in HGSOC, despite its name, originates from both the fallopian tube epithelium and the ovarian surface epithelium [7].

One of the major issues in uncovering the pathogenesis of the OC is the heterogeneous nature of the disease, comprising various histologic types (only 10% of OC are non-epithelial ones) with different behaviors and characteristics [8,9,10,11].

The four most common types of OC are actually epithelial ones: serous, clear-cell, endometrioid, and mucinous tumor: they are furtherly categorized into sub-types according to their biological features and response to treatment [12].

The latest WHO classification of female genitals cancer, recognize at least five main types of ovarian cancer: high-grade serous carcinoma (HGSC, 70%), endometrioid carcinoma (EC, 10%), clear cell carcinoma (CCC, 6–10%), low-grade serous carcinoma (LGSC, 5%) and mucinous carcinoma (MC, 3–4%) [13].

Previous classification identified two different subtypes of epithelial ovarian carcinogenesis, Type I and Type II, diverging in epidemiology, etiology, and therapeutic approach. Invasive epithelial OC (Type I), genetically stable and asymptomatic, is generally detectable in low-grade serous, endometrioid, clear cell and mucinous OCs: they typically proliferate slowly, can be diagnosed early and may have a good rate of prognosis.

Conversely, Type II OCs are classified as high-grade tumors and are commonly di-agnosed at advanced stages: they have a more aggressive progression and an extreme rate of proliferation. Moreover, Type II OCs are genetically unstable with p53 mutations pecu-liar to high-grade serous tumors [6]. Carcinosarcomas, serous and endometrioid G3 subtypes and undifferentiated OC are Type II ones: they’re responsible for 75% of OC morbidity, are identified generally in FIGO stages Ill or IV and are characterized by poor prognosis and early relapse [14].

This division of OC into two types has had an important role historically [12,13,15] but nowadays it’s gradually losing its relevance since there are new ways of approaching to OC subclassification, considering and integrating mutational and immunohistochemical data [16].

OC specific risk depends on many factors including age and genetic predisposition; it has been well assessed that Hereditary Ovarian Cancers (HOC) is related to germline mutations in the BRCA gene, revealing that genetic testing for BRCA1/2 mutation carriers is a key point in the risk evaluation and clinical management of OC [17,18]. Although it is known that some lower penetrance genes may represent additional cancer risk factors, further studies are required to fully define their role in OC development. It’s generally recognized that these rare mutations (less than 1% in the general population) are related to 20% of the extra-familial risk factor [19,20]. Moreover, ethnic backgrounds have also been revealed as genetic risk for the onset of the disease (i.e., Jewish, French Canadian, Dutch, and those of Icelandic descent) [21].

Several investigations described an increased risk of OC in women with a high number of ovulatory cycles, as a consequence of the pro-inflammatory response of the distal fallopian tubes that occurs during ovulation [22].

Twenty-one percent of epithelial ovarian cancers are linked to lifestyle and other risk factors, (such as alcohol, obesity, smoking, talc use, diet). Minor studies have shown a link between dietary fiber eating and its correlation with the prevalence of OC [23].

Furthermore, low levels of vitamin D have been related with an increased risk of this malignancy [23]. Whereas oral contraceptive pill (OCP) use, pregnancy, breastfeeding, and tubal ligation are well-established defensive factors [24].

Finally, also endometriosis could be considered a risk factor since endometriosis-associated epithelial OCs have been frequently observed in younger women, luckily with a better overall prognosis [25].

## 2. HE4 as a Multipurpose Biomarker in OC

Currently, OC is a target of intense biomarkers research because it is often diagnosed in late stages (III or IV) when the survival rate is less than 20%: efforts have therefore focused on diagnosing early-stage or low-volume disease through accurate tumor markers [26].

In this context, serum determination of the Human epididymis protein 4 (HE4) biomarker is one of the most promising early detection test [26]. At present there are several methods available to measure HE4 serum levels: Enzyme-linked immunosorbent assay (ELISA), Radioimmunoassay (RIA), Chemiluminescence immunoassay (CLIA) and Electrochemiluminescence immunoassay (ECLIA). Automated CLIA and ECLIA technologies have high reproducibility and are more sensitive compared to other tests [27].

HE4 is encoded as a protein of approximately 50 amino acids by the WFDC2 (whey acidic protein four-disulfide core domain protein 2) gene localized on chromosome 20q12-13.1. The Whey Acidic Protein domain, known also as WAP Signature motif, retains an evolutionarily highly conserved motif characterized by eight cysteines forming four disulfide bonds at the core of the protein that is peculiars to all family members [28].

HE4 is secreted extracellularly as glycoprotein into the bloodstream and acts as a protease inhibitor, inhibiting serine, aspartyl and cysteine proteases; however, its exact function remains unknown [29].

It was firstly identified as a transcript exclusively expressed in distal epididymis epithelium by researchers at the Pacific Northwestern Research Institute in Seattle. Due to its localization, it was proposed as a specific tissue marker for the same [29].

Later studies demonstrated that HE4 was expressed in several epithelia such as the oral cavity, respiratory tract, female genital tract, distal renal tubules and in the colonic mucosa but interestingly it was not found in normal ovarian surface epithelium [30]. The first report on the possible use of HE4 as a tumor marker in ovarian cancer was published by Hellstrom et al. in 2003 [31]. As mentioned before HE4 expression in ovarian tumors depends on the histological subtype: HE4 is expressed mainly in serous and endometrial ovarian tumors which constitute the majority of OCs, and rarely in mucinous epithelial and germinal cancers [27]. As a clinical biomarker, HE4 is considered an early detection method: some studies suggest increased serum HE4 expression in nearly 92% of patients with OC, showing similar sensitivity and increased specificity to CA125, the current gold standard biomarker of this malignancy [30]. Compared to the more established CA125, HE4 is in fact less frequently increased in benign gynecological conditions such as endometriosis, adenomyosis, uterine fibroids or even menstruation, and thus these do not compromise its specificity [32,33]. HE4 was also shown to have a higher sensitivity than CA125 in detecting early stages of OC [34]. In 2009, the US Food and Drug Administration (FDA) agreed to use HE4 to diagnose and monitor OC [35]. Later in 2011, it also approved its use in combination with the marker CA125 as a part of the Risk of Ovarian Malignancy Algorithm (ROMA) for the OC diagnosis, especially in the presence of adnexal masses [36]. ROMA is a logistic regression algorithm that combines HE4 and Ca125 values and patient’s menopausal status to categorize women into high- and low-risk chances that a tumour will be diagnosed in a female subject with a pelvic mass.

In spite of elevated sensitivity and negative predictive value (NPV) (respectively 94% and 99%) ROMA demonstrated a specificity of 75% for predicting OC in women with a pelvic mass: therefore the algorithm has been considered less reliable than the biomarker HE4 alone [37]. Some studies suggest increased serum HE4 expression in nearly 92% OC patients, showing similar sensitivity and increased specificity to the CA125 marker [37]. Most interestingly, HE4 plays a unique role in OC as it has been implicated not only in OC diagnosis, but it may have also prognostic and predictive potential: several studies investigated HE4 importance for OC early detection, follow-up period, remission monitoring and therapy response. HE4 is in fact potentially able to identify patients that are at high risk for primary platinum resistance [26,36]. High HE4 expression correlates closely with poor survival of OC and may display an important role in early prediction of OC recurrence when compared to CA125 [37,38]. HE4 combined with CA125 seems to have a more rigorous prognostic for malignancy than either alone and as it has been recently reported that combining both biomarkers rapidity is more effective than either marker solely [39]. Notably, a different expression of HE4 was reported between BRCA1/2 mutated and sporadic OC patients: high levels of HE4 were predominantly observed in wild type patients. Moreover, in these patients, HE4 increased levels have been correlated to micronodular carcinomatosis and to a poor prognosis [18,40]. Indeed, recent evidence focused on the HE4 involvement in cellular proliferation, tumor growth, metastatic ability, chemoresistance and immune response suppression [2,3,18,27,41]. Based on these observations, the main goal of this review is to clarify the role of HE4 as a key molecule in the oncogenetic processes underlying OC so that HE4 could be possibly considered as an essential resource not only for diagnosis but also for prognosis and therapy choice (Figure 1).

## 3. HE4 in Cell Proliferation and Tumor Growth

In the last decade HE4 has been the subject of extensive investigations given its importance in OC growth and proliferation. This evidence was strengthened by several studies, demonstrating that gene expression profiles were altered in response to HE4 overexpression [41,42].

HE4 has a pivotal role in several signaling pathways of OC pathogenesis, summarized in Table 1.

The importance of HE4 in tumor progression has been investigated in vitro and in vitro, affirming its role in promoting proliferation through cell cycle regulation [44]. To this regard, HE4 silencing experiments performed in cell line SKOV3 showed cell cycle arrest in G0/G1 phase, preventing the entry into S phase. In contrast, stimulation with recombinant HE4 promotes the switch to G2/M phase [42,55]. Furthermore, in vivo studies demonstrated that HE4 knockout mice show a decrease in OC growth [56].

Among them, some reports showed that HE4 silencing experiments performed in OC cell line SKOV3, downregulated ERK pathway, significantly reducing cell proliferation, whereas overexpression of HE4 leads to ERK activation [44,45,46].

Another mechanism in which HE4 is involved is EGF/EGFR pathways. The binding of EGF to its membrane-bound receptor EGFR, leads to activation of the MAPK/ERK pathway [41,44,47]. There are two mechanisms by which HE4 interacts with the EGF/EGFR pathway: direct one through the binding to EGFR; indirect following EGF treatment that increases HE4 expression and nuclear translocation [52]. To confirm these results, recent findings demonstrated that HE4 knockdown significantly lowered ERK and EGFR phosphorylation levels, inhibiting SKOV3 cell line growth [41,44].

PI3K/AKT signaling pathway is generally overexpressed during cancerogenesis, since AKT has been established as a strong tumorigenesis promoter [41]. Several studies showed that HE4 overexpression induced AKT increase promoting cell growth, whereas HE4 knockdown has a reverse effect [41,45].

The histone deacetylase 3 (HDAC3) and HE4 are both present in the nucleus and cytosol of cellular fraction and is known that HDAC3 promotes proliferation, invasion and migration in ovarian cancer cells. Indeed, it has been reported that the interaction between HDAC3 and HE4 activates the PI3K/AKT signaling pathway. Moreover, HE4 expression is directly affected by HDAC3 attendance [49].

## 4. HE4 and Angiogenesis

In the tumor microenvironment, HE4 has the potential to alter signaling pathways to modify the expression of related genes. Since HE4 is a secretory protein, it can function intracellularly via autocrine or paracrine mechanisms [57].

Tumor progression and angiogenesis are usually influenced by hypoxia, a condition mediated by hypoxia-inducible factor 1-alpha (HIF1α) that, when associated with hypoxic condition, can modulate VEGF, which is essential for neovascularization [58]. The angiogenic function of HE4 in OC is promoted by VEGF jointly with epidermal growth factor (EGF) and insulin: it has also been demonstrated that VEGF synthesis is promoted by IL-1α which is directly proportional to HE4 concentration [59,60,61]. Another evidence supporting the pivotal role of HE4 in OC oncogenesis is that it is able to activate STAT3 protein, a multifunctional transcription factor involved in multiple biological processes [54].

The activation of STAT3 mediated by HE4 promotes transcription of pro-angiogenic factors such as IL-8 and HIF1α: IL-8 leads to persistent neutrophil recruitment in tumor tissue stimulating neoangiogenesis whereas HE4-HIF1α interaction has to be better characterized yet [62,63].

HIF1α is a key mediator of cellular adaptation to hypoxia through the activation of signaling and metabolic pathways that promote cell survival [64,65]. In vitro experiments performed by inhibiting HIF1α function, showed a significant decrease in HE4 levels [41,51,52]. Moreover, the physical interaction between HIF1α and HE4 gene promoter regions increases HE4 protein synthesis [66,67,68].

The alteration of these pathways and in particular of IL8 and HIF1α is described in HGSOC as an unfavorable prognosis factor [69,70].

It has also been reported that STAT3 inhibitors are able to block HE4-mediated endothelium cavity formation by reducing IL-8 and HIF1α [71].

Supporting the role of HE4 in OC microvascular invasion, there are evidences claiming that HE4 serum levels are positively related to an increased microvascular density in OC tissue [57].

Tumor angiogenesis is also regulated by programmed cell death-1 (PD-1), which suppresses the antitumor function of CD8+ T cells [72].

HE4 has also been shown to be involved in the regulation of other proteins that are responsible for angiogenesis such as metalloproteases, AKT, and annexin II (ANXA2) [45,46,73,74]. As described in literature, HE4 and ANXA2 (a neoangiogenic inducing factor), were both CD147 interacting proteins: HE4 could promote the invasion and metastasis of OC by regulating the expression of CD147. Also, this mechanism has been suggested as an intriguing therapeutic target of OC [75].

## 5. HE4 and Metastatic ProcessI confirm

During neoplastic progression, tumor cells detach from the primary tumor to colonize a new organ and giving rise to a secondary tumor, also named metastasis (from the Greek μετάστασις, “displacement”). Metastasization is a process finely tuned by several interactions that generally occurs between tumor cells and surrounding microenvironment (Tumor Micro Environment-TME) [76].

The bi-directional crosstalk between tumor cells and surrounding TME can promote the acquisition of epithelial mesenchymal transition (EMT) phenotype consequently driving tumor progression. The concept of EMT is of high clinical significance as it is associated with higher tumor grade, tumor relapse, and increased metastasis.

In this scenario, HE4 role has been debated for a long time: previous studies on the functional role of HE4 in multiple cellular processes indicated a potential defensive role of HE4 in OC progression [77].

However, with the progress of molecular techniques, scientists reached a different point of view on the HE4 role in OC tumorigenesis [78].

In 2019 Wang et al. performing HE4 knockout experiments, defined several functions of this protein: they reported that the reduction of HE4 expression inhibits cell proliferation switching on the apoptotic way by activating caspase-3, cleaved caspase-3 and reverting EMT. This effect is confirmed by western blot analysis showing the increasing of E-cadherin protein and the reduction of N-cadherin and Snail levels [54]. Finally, the expression of metalloproteinases 2 and 9 is also inhibited by HE4 downregulation [78].

Most of these mechanisms are strictly connected with the JAK/STAT3 activation pathway which, in the absence of HE4, is strongly impaired. To this regard, HE4 knockdown experiments performed in vitro showed a suppression of the malignant progression by inhibiting the JAK/STAT3 pathway [54].

Further studies strengthened these findings demonstrating that HE4, due to its association with lymphatic metastasis and other unfavorable factors in endometrial cancer, is a useful preoperative prognostic marker [78].

Interestingly, in OC cells high levels of HE4 fucosylated antigens such as Lewis y antigen were detected. Studies performed on this antigen showed that when Lewis y antigen is highly expressed on HE4 surface, it ascribes a highly metastatic phenotype to OC cells [77,78,79]. The possibility to hide and block Lewis y antigen by using a specific antibody, could be a feasible method for the therapeutic treatment of OC [79].

## 6. HE4 and Immune Response

HE4 was initially suggested to have a potential role in innate immune defense of multiple epithelia, and recently few studies have been conducted to better understand HE4 function in tumor immunity [80].

Several lines of evidence suggest that during cancer proliferation, TME creates the conditions for which the immune system is no longer able to recognize and destroy cancer cells. In fact, during tumor progression, while the immune system tries to defend the body, the tumor produces molecules that reduce the effectiveness of the immune response [81].

Among them, the most attractive one is Programmed Cell Death Ligand 1 (PD-L1), a trans-membrane protein considered a co-inhibitory factor of the immune response. PD-L1 is a protein expressed at low levels in different cell types functioning as a modulator of excessive or unwanted immune responses. Recent studies reported that HE4 overexpression increases PD-L1 expression on both tumor cells and macrophages through a novel post transcriptional mechanism [82].

As discussed before, a positive correlation between HE4 serum levels and IL-8 has been described in patients with microvascular ovarian cancer, affecting trafficking of cytotoxic T lymphocyte; the trip of these cells into the tumor tissue is dysregulated and suppressed as a consequence of HE4 increase [83].

The biological meaning of HE4 overexpression has also been studied in OC tumor immune microenvironment by comparing syngeneic model of rat ovarian cancer with human patient data. In this model, it has been observed that in malignant ascites, HE4 overexpression supports M2 macrophages recruitment and at the same time reduces the recall of activated CTL and NK. To this regard, further studies highlighted a particularly aggressive molecular subtype of HGSOC, suggesting that patients with this histotype having a higher antigen-presenting cell infiltration and PD-1/PDL-1 expression, can benefit from a specific immunotherapy [82].

It has also been revealed a significant positive correlation between Regulatory T cells (Tregs) subpopulations and HE4: Tregs are usually enriched in OC, and their immunosuppressive function plays a key role in tumorigenesis and progression [84].

Additionally, some recent studies have shown that HE4 is involved in promoting ovarian tumor immune evasion, through influencing expression of two proteins, osteopontin (OPN) and dual specificity phosphatase 6 (DUSP6): consequently, through targeting of HE4, it may be possible to downregulate molecular mechanisms that promote tumorigenesis and to restore a normal tumor immune response too [85].

## 7. HE4 and OC Therapy

The first-line standard care of patients with OC is extensive debulking surgery combined with taxane compounds (such as paclitaxel) and platinum-based agents (such as carboplatin).

Use of frailty assessment tools to predict postoperative adverse outcomes and overall survival in frail patients with gynecological cancer.

It has been shown a negative predictive value of about 80% for HE4 for optimal cytoreductive surgery [86]. It has also been demonstrated that the combination of CA125, HE4, and computed tomography is more effective at predicting the presence of residual tumors post-neoadjuvant chemotherapy [87]. More than 25% of patients with relapse have a poor prognosis due to a platinum-resistant or platinum-refractory disease. Thus, angiogenic inhibitor (such as bevacizumab), immune checkpoint inhibitors, poly ADP-ribose polymerase (PARP) inhibitors, estrogen receptor inhibitors, and various inhibitors of intrinsic tumor signaling pathways are being used as potential therapeutic agents for OC patients [88].

HE4 and ROMA scores are more sensitive predictors of platinum response than CA125 alone [52].

Angioli et al. reported that serum HE4 levels predict platinum-resistant versus sensitive disease at the third chemotherapy cycle with 100 % sensitivity and 85 % specificity [89]. Moore et al. demonstrated that patients resistant to first-line chemotherapy have increased HE4 serum levels.

Furthermore, higher HE4 levels inversely correlate with clinical outcome, optimal cytoreduction and overall survival inversely correlate with increased HE4 levels: the latter can also be considered as an independent prognostic factor for progression free survival [37].

Additionally, HE4 overexpression enhances concomitant resistance to cisplatin and paclitaxel, and this chemoresistance can in part be reversed downregulating HE4: it can be speculated that small molecules or antibodies targeting HE4 may enhance the efficacy of first- or second-line therapeutics and reduce the development of resistance [46].

In an interesting study it has been revealed that in the *BRCA* WT patients, HE4 performed as a predictive marker of chemosensitivity with a sensibility of 80% and a specificity of 100%; in *BRCA* mutated women, HE4 performed as a predictive marker of chemosensitivity in all patients: the ability to detect platinum-resistant patients before tumor relapse probably could open new therapeutic scenarios [40].

Many successful OC regimens are combination therapies that produce higher response rates and lower resistance rates compared with monotherapies [90].

Numerous scientific studies are being conducted to obtain new methods of treatment that will enable better patient survival and at the same time there is a constant effort to find biomarkers as new therapeutic targets. With an increasing understanding of OC progression mechanisms, selective molecular targeted therapies are emerging as innovative and promising therapeutic strategies: in this novel scenario biomarkers assume greater importance in the clinical management of OC.

Taken together, all the findings reported in the previous paragraphs highlight the key role played by HE4 in OC progression and metastasis, thus suggesting this biomolecule as a novel therapeutic target for the malignancy.

Figure 2 Summarizes the HE4 interplays with different routes driving OC oncogenesis.

## 8. Conclusions

Diagnostic, prognostic and predictive biomarkers of elevated sensitivity and specificity could allow an early diagnosis, risk stratification and consequently a better survival rate for women affected by OC, currently the most frequently fatal gynecologic neoplasm [91].

Several circulating molecules have been identified as possible disease markers (i.e., osteoprotegerin, survivin, metalloproteinases) [92,93,94], however, considering the recent updates from literature, we can define HE4 not only as a diagnostic, prognostic and predictive circulating biomarker, but we can refer to HE4 as a biological agent effectively involved in many of OC tumorigenesis mechanisms. Indeed, there are several OC oncogenic pathways in which HE4 plays an important role: in the light of this new perspective, we can consider this protein as a valuable tool in aiding the current transition towards “personalized oncology”.

Nowadays, there is a constant effort in personalized therapeutic strategies: for example it has been speculated that assessing the frailty of patients suffering from gynecological malignant tumors and offering them a personalized therapeutic strategy could improve the oncological outcome; meanwhile, with regard to surgical treatment of early-stage OC, it has been suggested that implementing minimally invasive approach might reduce lymphatic complications after OC staging [95,96].

Taking into consideration advancements in precision therapeutic strategies, detecting clinically relevant predictive tumor biomarkers, and subsequently ideal candidates for these treatments, become fundamental [97].

Further studies better elucidating the exact mechanisms in which HE4 drives OC pathogenesis will ultimately provide evidence as to whether HE4 should be recommended as a therapeutic target for this lethal disease.

## Figures and Tables

**Figure 1 ijms-24-10479-f001:**
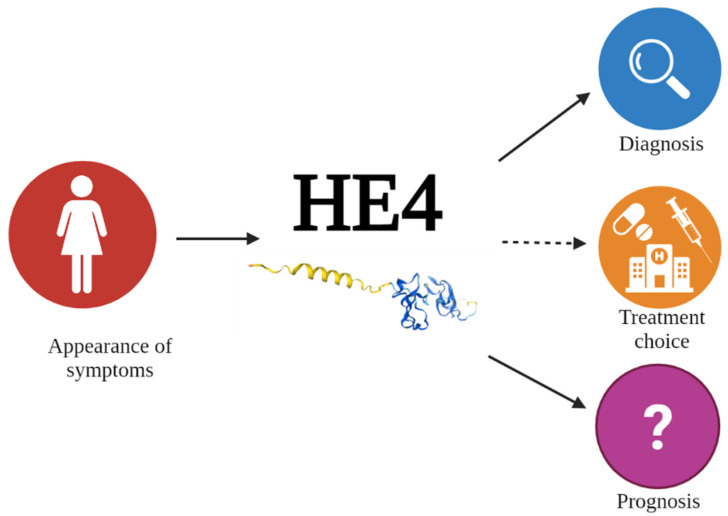
HE4 as a disease checkpoint. After the appearance of symptoms, HE4 is a biomarker of crucial importance since it can lead to a correct and tempestive diagnosis.

**Figure 2 ijms-24-10479-f002:**
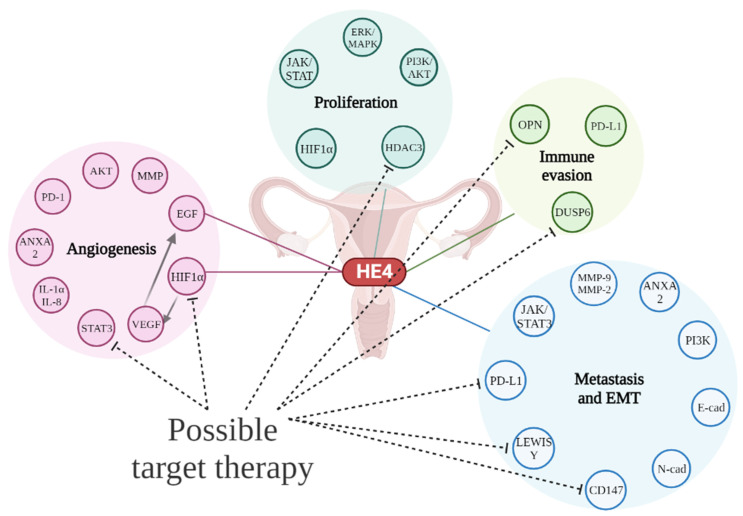
Schematic representation of HE4 interaction network in OC oncogenesis. HE4 is involved in cellular pathways of neoplastic proliferation and immunological evasion. Intracellular signaling and interaction with microenvironment can increase aggressive phenotypes that underlie angiogenesis and metastatic dissemination. Drugs that can interfere with each pathway shown in figure. STAT3, HIF 1α, HDAC3, OPN, DUSP6, LEWIS Y and CD147 may represent possible target of OC therapy (represented by dotted line), still under investigation.

**Table 1 ijms-24-10479-t001:** Pathways and functions linking HE4 to OC oncogenesis.

Pathway	Description and Functions	HE4 Influence
**ERK/MAPK** (extracellular signal-regulated kinases/ mitogen-activated protein kinase) [41,43,44,45,46,47]	pathway usually activated by EGF. phosphorylation of neighboring proteins by ERK (“on” or “off” switch) ERK required for activation of genes for entry into the cell cycle pathway mutated in many cancers	regulation of proliferation and invasion of SOC cells ERK activation with HE4 overexpression decrease in proliferation when HE4 was silenced in SKOV3 cells. activation of ERK/MAPK pathway following the interaction of HE4 with EGF/EGFR
**PI3K/AKT** (phosphoinositide 3- kinases/ Protein kinase B) [41,45,48]	PI3K indirectly activates AKT following the phosphorylation of phosphatidyl inositol 4,5 bisphosphate (PIP2) and phosphatidyl inositol 3,4,5 trisphosphate (PIP3) phosphorylation of protein substrates by AKT activation of biochemical pathways leading to cell growth and resistance to apoptosis mTOR protein involved in angiogenesis and increase of membrane glucose transporters	AKT increase promoting cell growth in OVCAR3 cells when HE4 is overexpressed. AKT decrease and subsequent reduced cell growth in OVCAR3 cells when HE4 is knockdown
**HDAC3** (histone deacetylase 3) [49]	role in S phase progression, DNA damage control, genomic stability maintenance	HDAC3 expression or knockdown lead to a corresponding increase or decrease in HE4 expression HE4 and HDAC3 binding activates the PI3K/AKT signaling pathway inhibition of the interaction between HDAC3 and HE4 may have potential therapeutic value
**HIF1α** (hypoxia-inducible factor 1-alpha) [41,50,51,52]	key mediator of cellular adaptation to hypoxia involved in processes of proliferation, survival and angiogenesis modulated by hydroxylation, acetylation, and phosphorylation	HE4-HIF1α interaction is yet not well understood decrease in HE4 levels in SKOV3 cells treated with HIF1α siRNA or with HIF1α inhibitors
**JAK/STAT** (Janus kinases/signal transducer and activator of transcription proteins) [53,54]	pathway activated by cytokines and growth factors intracellular response triggered by the action of the activated STAT proteins alteration of gene expression of proteins involved in proliferation, differentiation and apoptosis	HE4 knockdown inhibits the activity of the JAK/STAT3 pathway in vitro and in vivo HE4 knockdown suppresses cell proliferation and malignant progression of ovarian cancer

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
