# Peer review of "Recent Insight about HE4 Role in Ovarian Cancer Oncogenesis"

_ijms, 2023, doi:10.3390/ijms241310479_

Round 1

Reviewer 1 Report

I read with great interest the Manuscript titled " Recent insight about HE4 role in ovarian cancer oncogenesis" which falls within the aim of the Journal.

In my opinion, this topic analyzed is interesting enough to attract readers’ attention.  Although the manuscript can be considered already of good quality, I would suggest the following recommendations:

-       I suggest a round of language revision, in order to correct few typos and improve readability.

-       The authors should discuss the importance of a correct and complete study of patients with ovarian cancer, considering the various comorbidities and advanced age, and the impact that this pathology and various therapeutic strategies could have in these patients (I suggest authors to read and insert in references the following article: The role of preoperative frailty assessment in patients affected by gynecological cancer: a narrative review, Ottavia D’Oria, Tullio Golia D’Auge, Ermelinda Baiocco, Cristina Vincenzoni, Emanuela Mancini, Valentina Bruno, Benito Chiofalo, Rosanna Mancari, Riccardo Vizza, Giuseppe Cutillo, Andrea Giannini Vol. 34 (No. 2) 2022 June , 76-83 doi: 10.36129/jog.2022.34)

-       It would be interesting to discuss the possible complications related to various surgical techniques in the treatment of ovarian cancer. (In particular, I suggest this article to get deeper in the topic PMID: 28347880)

Because of these reasons, the article should be revised and completed. Considering all these points, I think it could be of interest to the readers and, in my opinion, it deserves the priority to be published after minor revisions.

 I suggest a round of language revision, in order to correct few typos and improve readability.

Author Response

REW. 1. I read with great interest the Manuscript titled " Recent insight about HE4 role in ovarian cancer oncogenesis" which falls within the aim of the Journal.

In my opinion, this topic analyzed is interesting enough to attract readers’ attention.  Although the manuscript can be considered already of good quality, I would suggest the following recommendations:

-       I suggest a round of language revision, in order to correct few typos and improve readability.

Ok it has been revised.

-       The authors should discuss the importance of a correct and complete study of patients with ovarian cancer, considering the various comorbidities and advanced age, and the impact that this pathology and various therapeutic strategies could have in these patients (I suggest authors to read and insert in references the following article: The role of preoperative frailty assessment in patients affected by gynecological cancer: a narrative review, Ottavia D’Oria, Tullio Golia D’Auge, Ermelinda Baiocco, Cristina Vincenzoni, Emanuela Mancini, Valentina Bruno, Benito Chiofalo, Rosanna Mancari, Riccardo Vizza, Giuseppe Cutillo, Andrea Giannini Vol. 34 (No. 2) 2022 June , 76-83 doi: 10.36129/jog.2022.34)

-       It would be interesting to discuss the possible complications related to various surgical techniques in the treatment of ovarian cancer. (In particular, I suggest this article to get deeper in the topic PMID: 28347880)

 Thank you for your comment, we added a sentence and the relative references. Pag 10 lane 374. Ref 95-96

Because of these reasons, the article should be revised and completed. Considering all these points, I think it could be of interest to the readers and, in my opinion, it deserves the priority to be published after minor revisions.

Comments on the Quality of English Language

 I suggest a round of language revision, in order to correct few typos and improve readability.

Reviewer 2 Report

The work entitled "Recent insight about HE4 role in ovarian cancer oncogenesis" by Emanuela Anastasi raises the extremely important problem of early and specific diagnosis of ovarian cancer. As emphasized by the authors, malignant ovarian tumors do not have characteristic clinical symptoms and are usually detected in late stages of clinical advancement when therapeutic options are limited There are still no sufficiently sensitive and specific laboratory markers for ovarian cancer. The Authors in the paper focus on the characteristics of HE4, but I believe that it is worth mentioning other already used or promising laboratory parameters useful in the diagnosis of ovarian cancer (osteoprotegerin, survivin, metalloproteinases and others). In my opinion, it is also worth discussing the histogenesis of ovarian cancer, its new classification and the differences between the various types of lesions in the introduction (article in Ginekol Pol. 2012, 83, 454-457). - expand paragraphs 49-57, quote here definitely more literature items. The role of HE4 in the etiopathogenesis, course of the disease and determining the prognosis of different histological types of ovarian cancer may vary.   I appreciate that the described protein has been discussed by the authors in such a multidimensional way, although I think that the paragraph on the importance of HE4 in metastasis needs to be expanded.   I really like the figures proposed by the Authors - they are simple, legible and make it easier to understand the text. Perhaps it would be possible to prepare also graphically illustrating Table 1.   The cited literature data are well selected, most of them to articles from the last 10 years.   After expanding the work with the aspects mentioned above - I recommend the work for publication and congratulate the authors on choosing an ambitious research topic  

 I am not qualified enough to assess the quality of English in this paper

Author Response

REW 2. The work entitled "Recent insight about HE4 role in ovarian cancer oncogenesis" by Emanuela Anastasi raises the extremely important problem of early and specific diagnosis of ovarian cancer. As emphasized by the authors, malignant ovarian tumors do not have characteristic clinical symptoms and are usually detected in late stages of clinical advancement when therapeutic options are limited There are still no sufficiently sensitive and specific laboratory markers for ovarian cancer. The Authors in the paper focus on the characteristics of HE4, but I believe that it is worth mentioning other already used or promising laboratory parameters useful in the diagnosis of ovarian cancer (osteoprotegerin, survivin, metalloproteinases and others). 

Thank you for pointing this out we changed the sentence and added references .Page 10 and line 366. Ref 92-94

In my opinion, it is also worth discussing the histogenesis of ovarian cancer, its new classification and the differences between the various types of lesions in the introduction (article in Ginekol Pol. 2012, 83, 454-457). - expand paragraphs 49-57, quote here definitely more literature items. 

Thank you for this observations. Regarding the histogenesis of ovarian cancer, we cited a very recent article as reference n. 7 (Y. Brown, S. Hua, and P. S. Tanwar, "Extracellular matrix in high-grade serous ovarian cancer: Advances in understanding of carcinogenesis and cancer biology," (in eng), Matrix Biol, vol. 118, pp. 16-46, Apr 2023, doi: 10.1016/j.matbio.2023.02.004), we have reconsidered the most recent classification, expanded paragraphs 49-57. We also quoted a total of 6 literature items: 6-12-13-14-15-16

The role of HE4 in the etiopathogenesis, course of the disease and determining the prognosis of different histological types of ovarian cancer may vary.   I appreciate that the described protein has been discussed by the authors in such a multidimensional way, although I think that the paragraph on the importance of HE4 in metastasis needs to be expanded.

Thanks for this suggestion, we have added a few sentences in the HE4 and metastasis paragraph trying not to be redundant with some topics that are also included in the HE4 in cell proliferation and tumor growth and HE4 and angiogenesis paragraphs. Furthermore, the mechanisms described are also summarized in Figure 2. Page 7 and lane 236, lane 243 and lane 261.

   I really like the figures proposed by the Authors - they are simple, legible and make it easier to understand the text. Perhaps it would be possible to prepare also graphically illustrating Table 1. 

Thank you for your suggestion we reformatted the Table1.

  The cited literature data are well selected, most of them to articles from the last 10 years.   After expanding the work with the aspects mentioned above - I recommend the work for publication and congratulate the authors on choosing an ambitious research topic  

Reviewer 3 Report

-        Language used could be improved to provide better clarity throughout the essay. There are instances of grammatical errors that obfuscate the meaning of sentences.

-        P.2 L.49: The specificity and clinical utility of TypeI/II classification is poor. The literature surrounding OC is migrating away from these classifications, which isn’t reflected in the paragraph provided herein. Consider rephrasing this section to reflect that the TypeI/II classification has had its role historically, but is losing relevance (arguably already irrelevant).

-        P.2 L.72: Citations required for this statement: “Minor studies have shown 72 a link between dietary fiber eating and its correlation with the prevalence of OC.”

-        Table 1 could have improved formatting and a more descriptive title (Summary table of *the* main pathways involved. ‘The’ here is not descriptive). Title for tables should come above table. References should be given for all the assertions made.

-        Good information provided on molecular mechanisms. Pathways and interactions could be better represented in a figure.

-        Language used could be improved to provide better clarity throughout the essay. There are instances of grammatical errors that obfuscate the meaning of sentences.

Author Response

REW 3.  Language used could be improved to provide better clarity throughout the essay. There are instances of grammatical errors that obfuscate the meaning of sentences.

-        “P.2 L.49: The specificity and clinical utility of TypeI/II classification is poor. The literature surrounding OC is migrating away from these classifications, which isn’t reflected in the paragraph provided herein. Consider rephrasing this section to reflect that the TypeI/II classification has had its role historically, but is losing relevance (arguably already irrelevant).”

Thanks for this comment, we have revised the paragraph following your suggestions. Even if it is from 2004, this classifications remains still important, as reported in these recent references: (12-13-15 in the manuscript)

This division of OC into two types has had an important role historically  but nowadays it’s gradually losing its relevance since there are new ways of approaching to OC subclassification, considering and integrating mutational and immunohistochemical data. Ref 16

-        P.2 L.72: Citations required for this statement: “Minor studies have shown 72 a link between dietary fiber eating and its correlation with the prevalence of OC.”

OK it has been done reference 23.

-        Table 1 could have improved formatting and a more descriptive title (Summary table of *the* main pathways involved. ‘The’ here is not descriptive). Title for tables should come above table. References should be given for all the assertions made.

In agreement with this observation and with reviewer n2 comments, the table has been re-formatted. The Table title has been changed,  moved above the table and references for all the assertions have been added.

-        Good information provided on molecular mechanisms. Pathways and interactions could be better represented in a figure.

Thank you for this  suggestion but since the network of molecular mechanism is very intricate and detailed explanation required a deep description we summarized   them  in fig 2. This could be an excellent starting point for a future review on the topic.

Comments on the Quality of English Language

-        Language used could be improved to provide better clarity throughout the essay. There are instances of grammatical errors that obfuscate the meaning of sentences.